# Living Alone but Not Feeling Lonely: The Effect of Self-Concealment on Perceived Social Support of Youth Living Alone in China

**DOI:** 10.3390/ijerph192113805

**Published:** 2022-10-24

**Authors:** Linran Zhang, Xiaoyue Fan, Zhanyu Yu

**Affiliations:** School of Education Science, Jiangsu Normal University, Xuzhou 221000, China

**Keywords:** youth living alone, self-concealment, psychological needs met through internet gratification, social self-esteem, perceived social support

## Abstract

The current study explored the mechanism of self-concealment on perceived social support among youth living alone and tried to clarify the two mediating variables, which are “psychological needs met through internet gratification” and “social self-esteem”, by using the Self-Concealment Scale, the Psychological Needs met through Internet Gratification Scale, the Texas Social Behavior Inventory and the Perceived Social Support Scale. Four hundred thirty-three working youth living alone who have lived alone or shared no emotional intersection with others were chosen as participants in this study. The results showed that: (1) the correlation between self-concealment, perceived social support, psychological needs met via internet gratification and social self-esteem was significant; (2) self-concealment positively predicted perceived social support; (3) self-concealment indirectly predicted perceived social support through the chain mediating effect of “psychological needs met via internet gratification” and “social self-esteem”. These results indicated that the self-concealment of youth living alone had a predictive effect on the perceived social support. The mechanisms of this effect included the direct effect of self-concealment and indirect effect through “psychological needs met via internet gratification” and “social self-esteem”.

## 1. What Is Known about This Topic, and What Does This Paper Add

Known: More and more young people live alone. They are lonely, single, addicted to the internet, and have poor living environments and fixed interpersonal circles.

Adds: 1. Youth living alone are not stereotypical images. This study uses quantitative research to prove with data that they live a good life, are not addicted to the internet, have good social contact, and can feel help and support when facing difficulties.

2. The active choice and enjoyment of living alone is a short-term stage in the life course of some individuals, during which they learn to take responsibility, constantly recharge and improve themselves, and enhance their ability to control their own life.

## 2. Introduction

Adolescence is the “jointing and booting stage” of life, and careful and sufficient attention, cultivation and guidance should be given to subjects at this stage [1]. Notably, growth in the economy delays the marriage age, resulting in a decrease in marriage rates, an increase in divorce rates, and an increase in the number of youth living alone. The psychological characteristics of this group should be explored. Youth living alone mainly refers to young people who have a certain degree of education, work in big cities, have self-respect, self-confidence, and independence, live alone or share a house with two or three people without any intersection with each other, and do not live with their parents and relatives. This group includes married and divorced people who still live alone [2,3,4].

Young people who have just graduated from college and worked in cities are most likely to live alone. According to a survey reported by the People’s Think Tank, the reasons for living alone are as follows: (i) they want their privacy; (ii) they have a different life schedule from others; (iii) they live alone easily; (iv) they passively chose to live alone because their roommate moved away [5]. In addition, the formation of youth living alone is influenced by peer groups and the environment [6]. According to previous studies, the situation of youth living alone is not optimistic, such as living in tiny houses, poor environments, high rent, and loneliness; after staying in big cities for a period, they usually interact with fellow natives, colleagues, or clients, so that their interpersonal relationship is a single and fixed form. At the same time, youth living alone may be subject to implicit discrimination at work, such as when they work overtime, they are taken for granted as the first candidates for overtime [6]. On the contrary, other studies hold a different view that, compared with family households, youth living alone have their own value and behavior style, conveying information and perceiving the world in the way that they understand [7]. Actually, they merely empty the living space, but not the social space [8]. A higher proportion of youth living alone is highly educated than those not living alone [9]. They have a strong sense of independence. Meanwhile, they also have more self-respect and self-confidence [8]. They believe that living alone is a necessary stage of life [10].

To sum up, most studies described the reasons and survival status of youth living alone, but few had in-depth studies and supporting data. This study uses quantitative research to explore the influence mechanism of self-concealment on perceived social support among youth living alone and the chain mediating effect of “meeting psychological needs through internet gratification” and “social self-esteem” so that individuals, families, and society can have access to an in-depth understanding of the status quo and characteristics about the social behavior of youth living alone. Furthermore, it can provide theoretical and empirical evidence for the intervention measures regarding the social behavior of youth living alone.

### 2.1. Relationship between Self-Concealment and Perceived Social Support

Perceived social support refers to the subjective judgment that an individual believes they can obtain respect, support, and understanding when they need help [11,12]. It is an important positive resource for the healthy development of children and adolescents [13]. Krinenberg reported that living alone does not always result in loneliness and may promote self-attention and self-reflection in youth living alone [14]. Notably, their attention to the outside world and others is relatively weakened. Moreover, Maslow reported that being alone is one of the characteristics of self-actualizers and can help in realizing the benefits of being alone, such as enjoying being alone and feeling relaxed and comfortable [15]. This indicates that the self-concealment level of youth living alone is high. Self-concealment refers to the psychological tendency of individuals to actively conceal personal information that may be negative or painful from other individuals. Self-concealment comprises three aspects: tendency to keep secrets to oneself, not sharing personal, painful secrets or negative thoughts about oneself and worrying about the leakage of concealed personal information [16].

Buchholz’s need theory indicates that being alone is a developmental need that helps individuals to perceive and regulate their negative emotions [17,18,19]. Further, the theory provides psychological and environmental space for self-healing and pain relief [20]. Living alone can also help in maintaining a good self-image, promoting harmony and stability of interpersonal relationships [21,22], and gaining the psychological feeling of being supported by others. Therefore, youth living alone actively choose to live independently and conceal adverse events. Youth living alone are more independent and have higher abilities of self-knowledge, self-evaluation, self-development, and self-realization [23]. Negative information from the outside and from other individuals has less impact on this group, and they prefer to solve personal problems by themselves and maintain privacy [24]. Therefore, youth living alone have a higher level of self-concealment and active choice and choose to actively solve the low-level pressure caused by living alone. Furthermore, youth living alone have more natural advantages of being alone. They express a better sense of control over the environment, experience happiness, relaxation, freedom, and optimism by living alone [25]. These positive emotions promote the feeling of being supported in this group. Therefore, self-concealment of youth living alone may positively predict perceived social support (Hypothesis 1).

### 2.2. Mediating Role of Psychological Needs Met through Internet Gratification

Youth currently living alone comprises a generation living in the internet era. The internet is an indispensable part of the lives of youth living alone [26]. Use of the internet mainly satisfies people’s need for emotional catharsis, social communication, entertainment, and professional status identification and other needs [27,28,29,30]. Psychological needs met through internet gratification refers to the satisfaction of specific psychological needs of individuals through the internet [31]. Self-concealment is manifested in protecting one’s privacy and suppressing negative emotions and other aspects, which can satisfy the individual’s pursuit of independence in real life [32]. In China, youth living alone pay attention to the relative independence of their inner world, carry out appropriate emotional expression in interpersonal interaction and are more willing to maintain a sense of boundaries with the outside world [33]. In addition, the social-emotional needs of the internet emphasize the clustering of individuals or groups, rather than independence [34]. Therefore, self-concealment can negatively predict psychological needs (internet gratification).

Wang et al., reported that people with a lower degree of psychological need satisfaction are more likely to use the internet to adjust social emotions, thus obtaining a sense of support [35]. Previous studies evaluated the predictive effect of perceived social support on meeting psychological needs through internet gratification [36,37]. Moreover, the Anonymity–Convenience–Escapism (ACE) model proposes that the internet can satisfy the need for individual anonymity, convenience, and escape from reality [38]. This indicates that individuals who spend more time and energy in the online world spend less time and energy in the real world. In addition, these individuals experience less social support, implying that meeting psychological needs through internet gratification negatively predicts perceived social support. In summary, meeting psychological needs through internet gratification may mediate the relationship between self-concealment of youth living alone and perceived social support (Hypothesis 2).

### 2.3. Mediating Role of Social Self-Esteem

Previous studies indicate that self-protection is a defensive strategy, and self-protection can help in avoiding, reducing and repairing negative self-perceptions caused by evaluations from other individuals [39]. The sense of self-worth and social competence of an individual in social interactions are referred to as social self-esteem [40,41]. This indicates that the self-worth and social competence of individuals increase when they use self-protection strategies to conceal themselves in social interactions. When individuals do not use this strategy, they are likelier to feel out of self-control and have low social efficacy. Therefore, self-concealment can positively predict social self-esteem. Meanwhile, individuals with higher levels of social self-esteem have a more robust ability to perceive social support from others [42]. Notably, social self-esteem is significantly positively correlated with perceived social support [43]. Based on this, social self-esteem may be a mediator in the relationship between the self-concealment of youth living alone and perceived social support (Hypothesis 3).

### 2.4. Chain Mediating Effect of Meeting Psychological Needs through Internet Gratification and Social Self-Esteem

The Displacing Social Activity Hypothesis states that the time spent by individuals using the internet replaces the time spent on real social activities [44]. Similar to other negative non-social entertainment activities, the internet provides personal entertainment activities, and people experience a certain sense of pleasure from this type of entertainment. These entertainment activities can result in withdrawal from social relationships and reduce the personal sense of self-worth in social communication. This indicates that meeting psychological needs through internet gratification may be negatively correlated with social self-esteem. In summary, “meeting psychological needs through internet gratification” and “social self-esteem” may mediate the relationship between self-concealment of youth living alone and perceived social support (Hypothesis 4).

### 2.5. The Current Study

These theoretical and empirical premises imply that self-concealment positively predicts perceived social support (and all its domains) (H1). Further, they indicate that the relationship is mediated by meeting psychological needs through internet gratification (H2) and social self-esteem (H3) as mediators operating individually (separately) and in a serial manner (H4) (see Figure 1).

The hypothesized serial mediation model assumes that self-concealment is associated with a lower level of meeting psychological needs through internet gratification, which contributes to decreased social self-esteem, which, in turn, is associated with higher perceived social support.

## 3. Methods

### 3.1. Participants and Procedure

A questionnaire survey was conducted on employed participants who lived alone or shared a house but had no emotional intersection. A total of 34 invalid questionnaires that contained missing data or were not completely filled were eliminated. Notably, 433 valid questionnaires were obtained, with an effective rate of 92.7%. The recruitment procedure for participants included 3 steps. Firstly, the Questionnaire Star software was used for online recruitment, and 78 eligible participants were recruited. Then, the purposive sampling method was used for re-recruitment, and 135 eligible participants were recruited. Finally, based on purposive sampling, the snowball sampling was used, and 180 eligible participants were recruited. The subjects included in this study comprised 266 males (61.4%) and 167 females (38.6%) and had an average age of 28.32 ± 3.087 years (age range 18–40 years), comprising 75 aged 18–25 (17.3%), 343 aged 25–33 (79.2%), 15 aged 33–40 (3.5%). Of these, 27 (6.2%) were only children, and 406 (93.8%) were non-only children. Their educational levels were 48 (11.1%) high school or below, 198 (45.7%) junior college, 175 (40.4%) Bachelor’s degree, 11 (2.5%) Master’s degree, and 1 (0.2%) Doctor’s degree. Of all the participants, 37 (8.5%) worked for 0–3 years, 143 (33.0%) worked for 3–5 years, and 253 (58.4%) worked for more than 5 years. Regarding marital status, 329 (76.0%) were unmarried, 103 (23.8%) were married, and 1 (0.2%) was divorced.

### 3.2. Measures

The Self-concealment Scale compiled by Larson and Chastain is composed of 10 items [16]. An example of the items is: “If I shared all my secrets with my friends, they’d like me less”. The items have a score from 1 (never) to 5 (always). A high score indicated a deeper degree of self-concealment. Cronbach’s alpha coefficient for the entire scale in the current study was 0.88.

The Psychological Needs Met Trough Internet Gratification Scale compiled by Wan et al., is composed of 44 items (such as “Get rid of loneliness and loneliness in the online world”), including eight dimensions, as follows: influence (8 items), self-identification (7 items), meeting challenges (3 items), interpersonal communication (8 items), evasion of reality (6 items), autonomy (3 items), cognition (5 items), and achievement (4 items). The items are rated from 1 (strongly disagree) to 4 (strongly agree) [31]. A high score indicates a high degree of satisfaction of online basic psychological needs. Cronbach’s alpha coefficient in the present sample was 0.94.

The Texas Social Behavior Inventory revised by Helmreich and Stapp is composed of 16 items with scores from 1 (strongly disagree) to 5 (strongly agree) [45]. An example of the items is: “When I disagree with others, my opinion often prevails”. A high score indicates high social self-esteem. In the current study, the complete set of 16 items yielded an alpha coefficient of 0.85.

The Perceived Social Support Scale compiled by Jiang is composed of 12 items, including family support (4 items), friend support (4 items), and other support dimensions (4 items) [46]. An example of the items is “There are people (bosses, relatives and colleagues) who will be there when I have problems”. The items are rated from 1 (strongly disagree) to 7 (strongly agree). A high score indicates that the individual’s perceived social support is high. Cronbach’s alpha of this scale in the present study was 0.87.

### 3.3. Data Analysis

Statistical analyses were performed using IBM SPSS Statistics 26.0 (IBM, Armonk, NY, USA). Preliminary analyses included a determination of descriptive statistics of major variables (the means, standard deviations) and Pearson’s correlation analysis and a test of the possibility of common method deviation in the survey data. Substantive serial mediation analysis included bootstrap analysis and calculation of fit indices using the Mplus 7.4 tool. A saturated model is a condition on the assumption that all the parameters to be estimated in the model are exactly equal to the elements in the covariance matrix, the degree of freedom of the saturated model is 0, the chi-square values are also equal to 0, and the saturated model is called just identified, which leads to perfect fitting, so the fitting index is no longer estimated [47,48,49]. Let us just focus on the path coefficients. We used chi-square values (χ2/df), the comparative fit index (CFI), the Tucker–Lewis fit index (TLI), the root mean square error of approximation (RMSEA), and the standardized root mean square residual (SRMR) to evaluate the models. In general, the saturated model fit is indicated by CFI, and TLI is 1 and RMSEA and SRMR is 0 [50].

## 4. Results

### 4.1. Common Method Deviation Test

Harman’s single factor method was used to explore the common method bias in the present study. The results showed a total of 12 factors with characteristic roots above one. The first component explained 28.19% of the total variance variation, and the critical measurement value was less than 40%, indicating that there was no common method deviation problem in the present sample.

### 4.2. Preliminary Analysis

Compared to females, males scored higher on average in all variables. Independent sample *t*-tests revealed no significant differences between the sexes regarding self-concealment, meeting psychological needs through internet gratification, and perceived social support. A significant sex difference was found regarding social self-esteem (Table 1).

ANOVAs showed no significant interaction effect between gender and age group on the variables, i.e., self-concealment, meeting psychological needs through internet gratification, social self-esteem, and perceived social support. The sample’s main effect analysis showed that: (i) self-concealment: where youth living alone aged 25–33 years (3.334 ± 0.537), reported higher self-concealment than 18–25 years (3.049 ± 0.736) (*p* < 0.001); (ii) psychological needs met through internet gratification: where youth living alone aged 25–33 years (2.307 ± 0.289) had a fewer psychological needs met through internet gratification, compared to those aged 18–25 years (2.387 ± 0.432) (*p* < 0.05) and 33–40 years (2.327 ± 0.321) (*p* < 0.05); (iii) social self-esteem: in which youth living alone aged 25–33 years (3.186 ± 0.248) reported higher social self-esteem than those aged 18–25 years (3.075 ± 0.347) (*p* < 0.001) (Table 2).

### 4.3. Descriptive Statistics and Correlations

Means, standard deviations, and correlations for the variables are presented in Table 3. Correlation analysis showed that self-concealment was significantly negatively correlated with psychological needs met through internet gratification. Further, self-concealment was significantly positively correlated with social self-esteem and perceived social support. Meeting psychological needs through internet gratification was significantly negatively correlated with social self-esteem and perceived social support. Moreover, social self-esteem was significantly positively correlated with perceived social support.

### 4.4. Serial Mediation Analyses

The Mplus 7.4 tool was used to perform structural equation modeling for analysis of the association of self-concealment of youth living alone with perceived social support [51]. The direct effect of self-concealment of youth living alone on perceived social support was analyzed. The results showed that self-concealment significantly positively predicted perceived social support (γ = 0.331, t = 3.704, *p* < 0.001). Further analysis was performed to explore whether “psychological needs met through internet gratification” and “social self-esteem” mediated the relationship between self-concealment and perceived social support (see Figure 2). Notably, the model was a saturated model; therefore, the fitting index was no longer estimated, and only the path coefficients were relevant [47]. The results showed that self-concealment of youth living alone negatively predicted psychological needs met through internet gratification (γ = −0.368, t = −3.404, *p* < 0.001), and psychological needs met through internet gratification negatively predicted perceived social support (γ = −0.264, t = −2.745, *p* < 0.01). This indicates that meeting psychological needs through internet gratification mediates the effect of self-concealment on the perceived social support for youth living alone. Self-concealment of youth living alone positively predicted social self-esteem (γ = 0.335, t = 3.727, *p* < 0.001), and social self-esteem positively predicted perceived social support (γ = 0.514, t = 6.579, *p* < 0.001). This finding implies that social self-esteem mediates the effect of self-concealment on perceived social support of youth living alone. Psychological needs met through internet gratification negatively predicted social self-esteem (γ = −0.303, t = −3.098, *p* < 0.01). Notably, the direct effect of self-concealment of youth living alone on perceived social support was not significant (γ = 0.004, t = 0.047, *p* > 0.05). In summary, “psychological needs met through internet gratification” and “social self-esteem” play an intermediary role between self-concealment and perceived social support of youth living alone.

The bootstrap (1000 repeated samples) confidence interval method was used to explore the significance of the mediating effect (see Table 4). The direct path in the model was self-concealment → perceived social support. The 95% confidence interval was 0, and the effective value was 0.004. This indicates that the immediate effect of self-concealment on perceived social support was not significant. Further, three indirect paths were present in the model: (1) Self-concealment → psychological needs met via internet gratification → perceived social support. The 95% confidence interval for this path was not 0, and the effect value was 0.097, indicating a significant mediating effect. Therefore, ‘psychological needs met through internet gratification’ was a mediating variable between self-concealment and perceived social support, and H2 is valid. (2) Self-concealment → social self-esteem → perceived social support. The 95% confidence interval of the second indirect path was not 0 and the effective value was 0.172, indicating a significant mediating effect. This implies that social self-esteem was a mediating variable between self-concealment and perceived social support. Therefore, H3 was accepted. (3) Self-concealment → psychological needs met through internet gratification → social self-esteem → perceived social support. The 95% confidence interval of the third indirect path was not 0, and the effective value was 0.057, implying that the mediating effect was significant. This indicates that “psychological needs met through internet gratification” and “social self-esteem” both mediated the association between self-concealment serially and perceived social support among youth living alone.

## 5. Discussion

The present study examined gender and age differences in self-concealment, psychological needs met through internet gratification, social self-esteem, and perceived social support among youth living alone and sought to explore the relationship between self-concealment and perceived social support of youth living alone. The results from mediation analyses provided information on potential mechanisms that explain the relationship between self-concealment and perceived social support. Three types of mediation effects were observed, with “psychological needs met through internet gratification” and “social self-esteem” as separate single mediators and these variables playing a role as serial mediators.

### 5.1. The Difference in Age and Gender

This study found that males living alone had significantly higher levels of social self-esteem than females, which was consistent with previous studies [52]. The result showed that under the influence of traditional Chinese culture, males paid more attention to social ties than females and pursued personal success and the realization of self-worth [53].

Regarding age, the degree of self-concealment of youth living alone aged 25–33 was significantly higher than that of 18–25 years old. The possible reason for this result is that youth living alone aged 25–33 have a higher level of social cognition and pay more attention to self-growth. The negative information of the outside world has less influence on them, so they are more inclined to self-concealment. The level of psychological needs met through internet gratification of youth living alone aged 25–33 was significantly lower than those aged 18–25 and 33–40. This result showed that, during the critical period of work and career growth, youth living alone were more concerned about their development and did not rely too much on the internet for emotional expression or escapism. In terms of social self-esteem, only youth living alone aged 18–25 scored significantly lower than those aged 25–33, which may be due to the fact that with the accumulation of work practice and life experience, youth living alone had improved their sense of self-worth in the social process, their social ability had reached a certain level, and they may be in a more satisfactory state of social self-esteem.

### 5.2. Direct Effect of Self-Concealment on Perceived Social Support among Youth Living Alone

The findings of the current study showed that the self-concealment of youth living alone significantly positively predicted perceived social support. This implies that youth with high self-concealment perceived more social support. Youth living alone actively conceal some information about themselves owing to the accumulation of work experience and the complexity of interpersonal communication. This is an interpersonal communication strategy [54]. However, a previous study reported that self-concealment in high school students significantly negatively predicted their perceived social support [55]. In addition, previous findings showed that self-concealment in college students significantly negatively predicted social support [56]. The reason for the inconsistent results may be that the subjects of previous studies were mainly high school students and college students but not youth living alone. Youth living alone are employed and financially independent. Therefore, their social cognition level and experience are higher compared with those of high school and college students.

Costas and Grey reported that collective experience can be generated when people jointly conceal secrets, thus promoting the formation of small groups with high cohesion [57]. Individuals can actively hide and develop their interactions and work content freely. Therefore, youth living alone conceal some negative information and negative emotions that they think are unnecessary and have nothing to do with others. The youth should face work and communication in a relatively positive and friendly state and are likely to effectively experience the social support from groups living alone.

### 5.3. Mediating Role of Psychological Needs Met through Internet Gratification

Psychological needs met through internet gratification mediates the relationship between self-concealment and perceived social support among youth living alone. This implies that self-concealment of youth living alone reduces the level of psychological needs met through internet gratification, thus intensifying the level of perceived social support. The hyper-interpersonal model proposed by Walther indicates that individuals perform hyper-interpersonal communication in online interpersonal communication, which is different from general face-to-face interpersonal communication [58]. The sender and receiver of information selectively present themselves in the process of hyper-interpersonal communication. In this case, the sender and receiver have more opportunities to use interpersonal communication strategies, imagine and edit themselves, and selectively present themselves. Therefore, the level of psychological needs met through internet gratification may be lower because the youth comprise solitary groups with high independence [59], limited by the editability and selectivity of hyper-interpersonal communication. Moreover, studies indicate that self-concealment is negatively correlated with the satisfaction of psychological needs [60].

A previous study reports that basic psychological need satisfaction is significantly positively correlated with perceived social support [61]. However, the study mainly focused on the satisfaction of basic psychological needs in real life and ignored the difference between the satisfaction of psychological needs met on the internet and in reality. Individuals with low satisfaction in real life can benefit from online satisfaction through the satisfaction of network needs. On the contrary, individuals with low essential needs satisfaction in daily life cannot benefit from satisfaction gained online [62]. Although individuals flexibly switch between reality and online psychological needs [63], according to the ACE model, online and reality psychological need satisfaction exhibit an inverse relationship of time competition. Therefore, “psychological needs met through internet gratification” significantly negatively affects perceived social support.

### 5.4. Mediating Role of Social Self-Esteem

The relationship between self-concealment and perceived social support is mediated by social self-esteem among youth living alone. This implies that a high degree of self-concealment of youth living alone is correlated with a stronger sense of self-worth and social skills in social interactions, thus making it easier for perceived social support. The results of the present study support the theory of self-protection. Kahn reported that self-concealment can reflect the desire of individuals to express themselves in a communicative way that matches social expectations [64]. A previous study showed that Americans who lived alone spent more time with friends and neighbors than married people [65]. Youth living alone maintain a positive self-image through self-concealment, thus avoiding arousing resentment in interpersonal communication [66]. However, it helps in protecting their identity and reputation from being threatened or harmed [67]. Notably, social self-esteem was significantly positively correlated with perceived social support. This indicates that extensive social participation and frequent social interaction enhance the sense of self-worth in the group and the level of individual experience of social support, which is consistent with findings from previous studies [42]. Moreover, the “Use It or Lose It” hypothesis indicates that frequent social activities increase the cognitive activity of the individual and increase the sensitivity to social feedback. This sensitivity gradually decreases with the absence of social activities [68,69]. Therefore, social self-esteem plays a mediating role in the effect of self-concealment on perceived social support.

### 5.5. Chain Mediating Effect of Psychological Needs Met through Internet Gratification and Social Self-Esteem

The results of the chain mediation model showed that the degree of self-concealment affects perceived social support in youth living alone through the chain mediation of “psychological needs met through internet gratification” and “social self-esteem”. Notably, the level of perceived social support by youth living alone is affected by the degree of psychological needs met through internet gratification and by the sense of value and social skills of social interactions in the real world. However, the suppositional social activity hypothesis indicates that online and real social activities have an inverse relationship in time competition. Therefore, “psychological needs met through internet gratification” is negatively correlated with “social self-esteem”. The perceived social support of youth living alone is affected by the combined effect of the natural world and the online world.

Nicol explored the self-determined theory and reported that active solitude can satisfy advanced needs such as self-growth, self-recovery, independent thinking, discovery, and creation [70]. Moreover, solitude does not conflict with interpersonal skills [15,25,70]. Therefore, the results from the present study indicate that the social group of youth living alone should be treated more objectively and neutrally. Youth living alone are likely to conceal themselves to take control of their lives independently, materially and spiritually [71]. However, youth living alone are not addicted to the internet and have several psychological needs to be satisfied. Therefore, they actively participate in actual social activities and actively obtain social support through social self-esteem. The object-relations theory indicates that when individuals are actively alone, they can have a dialogue with their inner self and relieve negative emotions, such as anxiety, tension, and other negative emotions [72]. This finding shows that the solitary group conducts self-remodeling and self-improvement [26] and has better control of their own lives [73], rather than being in a lonely state of ‘no one asks me the temperature of porridge, no one stands with me at dusk’ [8,74].

Youth living alone may encounter problems in dealing with insufficient urban public space development, recessive social rejection, stereotypes, and discrimination, such as the labeling effect of adverse reactions [75]. However, the fact is that youth living alone were not confined to the surface of this sadness. It is a short-term option for a personal journey. More than anything else, they are willing to enjoy living alone and learn responsibility. They can arrange their time wisely to make constant progress and improve themselves. Enduring solitude allows them to gain control over their lives to pursue their dreams with all their might [10].

### 5.6. Limitations and Future Directions

There are several limitations of this study that should be noted. 

First, although we have made a great effort to collect sample data, future research should focus on youth living alone and expand the sample size to explore differences in psychological characteristics between youth living alone and other groups. Furthermore, the quantitative study is not enough to draw the conclusion of causality between self-concealment and perceived social support among youth living alone. In the future, the typical psychological aspects of this group should be evaluated through qualitative analysis, and studies should explore the influencing factors, development stages, and mechanism of social support among youth living alone. Last but not least, due to the Chinese culture, such as collectivism, marriage, and lineage, youth living alone in China are likely to be stigmatized. In this regard, relevant questionnaires or implicit association tests can be used for cross-cultural studies in the future.

## 6. Conclusions


(1)In terms of gender, there is a significant difference in self-concealment among youth living alone, and male is higher than female. In terms of age, there are significant differences in self-concealment, psychological needs met through internet gratification, and social self-esteem, but there is no significant difference in perceived social support.(2)Self-concealment can directly predict the perceived social support of youth living alone, and there is a significant positive correlation between them.(3)Self-concealment can also indirectly and positively predict perceived social support of youth living alone through the mediating effects of psychological needs met through internet gratification, and social self-esteem. Psychological needs met through internet gratification and social self-esteem can separately affect this relationship, but they can also influence this relationship as linked mediators.


## Figures and Tables

**Figure 1 ijerph-19-13805-f001:**
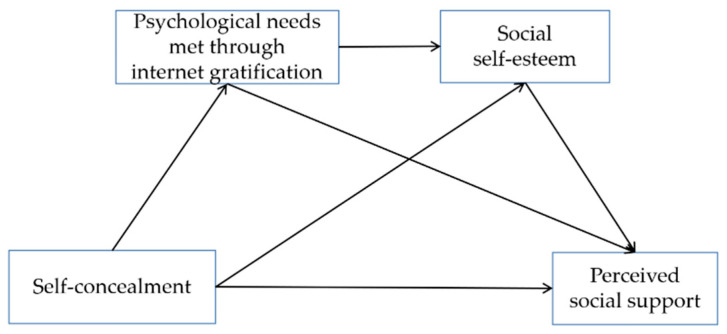
The hypothesized serial mediation model with ‘psychological needs met through internet gratification’ and ‘social self-esteem’ as mediators of the linkage between ‘self-concealment’ and ‘perceived social support’.

**Figure 2 ijerph-19-13805-f002:**
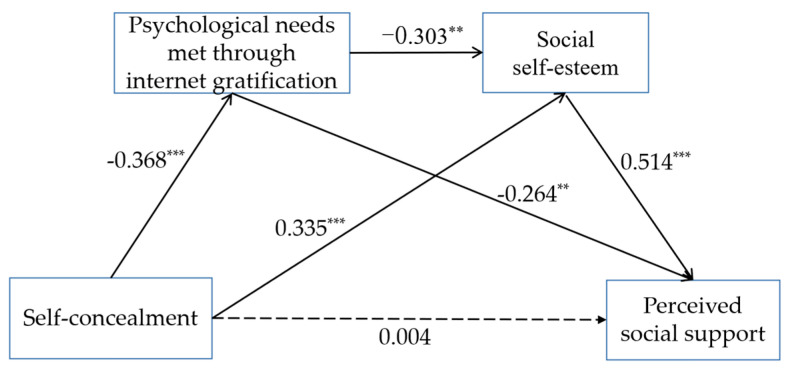
The serial mediation model with ‘psychological needs met through internet gratification’ and social self-esteem’ as mediators of the linkage between ‘self-concealment’ and ‘perceived social support’. Note: ** *p* < 0.01, *** *p* < 0.001.

**Table 1 ijerph-19-13805-t001:** Gender differences in all variables.

	Gender	Number	M ± SD	t	*p*
Self-concealment	male	266	3.306 ± 0.560	1.348	0.178
	female	167	3.228 ± 0.624		
Psychological needs met through internet gratification	male	266	2.328 ± 0.309	0.062	0.951
	female	167	2.326 ± 0.341		
Social self-esteem	male	266	3.193 ± 0.256	2.823	0.005
	female	167	3.119 ± 0.286		
Perceived social support	male	266	4.553 ± 0.648	0.685	0.493
	female	167	4.508 ± 0.697		

**Table 2 ijerph-19-13805-t002:** Age group differences in all variables.

	Age Groups (Years)	Number	M ± SD	F	*p*
Self-concealment	① 18–25	75	3.049 ± 0.736	8.342	0.001
	② 25–33	343	3.334 ± 0.537		
	③ 33–40	15	3.087 ± 0.541		
Psychological needs met through internet gratification	① 18–25	75	2.387 ± 0.432	4.059	0.018
	② 25–33	343	2.307 ± 0.289		
	③ 33–40	15	2.327 ± 0.321		
Social self-esteem	① 18–25	75	3.075 ± 0.347	5.507	0.004
	② 25–33	343	3.186 ± 0.248		
	③ 33–40	15	3.121 ± 0.244		
Perceived social support	① 18–25	75	4.478 ± 0.877	1.900	0.151
	② 25–33	343	4.560 ± 0.608		
	③ 33–40	15	4.250 ± 0.709		

**Table 3 ijerph-19-13805-t003:** Descriptive statistics and correlations among the study variables.

	M ± SD	1	2	3	4
1. Self-concealment	3.28 ± 0.59	—			
2. Psychological needs met through internet gratification	2.33 ± 0.32	−0.368 ***	—		
3. Social self-esteem	3.16 ± 0.27	0.447 ***	−0.426 ***	—	
4. Perceived social support	4.54 ± 0.67	0.331 ***	−0.485 ***	0.628 ***	—

Note: *n* = 433, *** *p* < 0.001.

**Table 4 ijerph-19-13805-t004:** Bootstrap analysis of the significance test of the intermediary effect of the chain intermediary model.

	Effect	Boot SE	Bootstrap 1000 Times 95% CI				Percentage
			Percentile		Bias Corrected		
			Boot LLCI	Boot ULCI	Boot LLCI	Boot ULCI	
Indirect 1	0.097	0.050	0.019	0.209	0.023	0.220	29.40%
Indirect 2	0.172	0.048	0.078	0.269	0.086	0.276	52.12%
Indirect 3	0.057	0.030	0.014	0.123	0.015	0.132	17.27%
Total Indirect Effect	0.326	0.075	0.188	0.482	0.194	0.490	98.79%
Direct	0.004	0.086	−0.157	0.178	−0.171	0.162	1.21%
Total Effect	0.330	0.089	0.155	0.502	0.146	0.496	

## Data Availability

The datasets generated for this study are available on request to the corresponding author.

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
