# Peer review of "Living Alone but Not Feeling Lonely: The Effect of Self-Concealment on Perceived Social Support of Youth Living Alone in China"

_ijerph, 2022, doi:10.3390/ijerph192113805_

Round 1

Reviewer 1 Report

Thank you for allowing me to review this manuscript. I think that the manuscript covered a timely topic. 

The study explored the self-concealment on perceived social support among youth living alone and focused on two mediating variables: psychological needs internet gratification and social self-esteem. 

Overall, the analyses are fairly appropriate for this type of work and the Introduction is straightforward. 

Few concerns: 

- The Authors should discuss the recruitment procedure for participants. 

- The authors should specify the number of items for each scale of measures used (for example, for the 8 scales of the "Psychological Needs Internet Gratification Scale")

- Authors should provide more information in the Data analysis section and specify all the descriptive analyses and the model used. 

Authors can move in this section some sentences used in the results section related to the justification of the test used.

- I propose to write not only a discussion but also conclusions and future research directions.

Author Response

Reviewer 1

The study explored the self-concealment on perceived social support among youth living alone and focused on two mediating variables: psychological needs internet gratification and social self-esteem.

Overall, the analyses are fairly appropriate for this type of work and the Introduction is straightforward.

Response:

Thank you very much for your affirmation of our hard work. Your opinions let us feel your love for academic research and meticulous research attitude again.

  1. The Authors should discuss the recruitment procedure for participants.

Response:

Thank you very much for the detailed comments.

Firstly,the Questionnaire Star software was used for online recruitment, and 78 eligible participants were recruited. Then, the purposive sampling method was used for re-recruitment, and 135 eligible participants were recruited. Finally, based on purposive sampling, the snowball sampling was used, and 180 eligible participants were recruited. 【197-201】

  1. The authors should specify the number of items for each scale of measures used (for example, for the 8 scales of the "Psychological Needs Internet Gratification Scale")

Response:

Thank you so much for the detailed comments.

The Psychological Needs Internet Gratification Scale including eight dimensions as follows: influence (8 items), self-identification (7 items), meeting challenges (3 items), interpersonal communication (8 items), evasion of reality (6 items), autonomy (3 items), cognition (5 items), and achievement (4 items). 【219-222】

The Perceived Social Support Scale including family support (4 items), friend support (4 items), and other support dimensions (4 items). 【232-233】

The Self-concealment Scale and the Texas Social Behavior Inventory are dimensional questionnaires.

  1. Authors should provide more information in the Data analysis section and specify all the descriptive analyses and the model used.

Response:

Thank you very much for the detailed comments.

Based on your comments, the authors added more information in the data analysis.

The subjects included in this study comprised 266 males (61.4%) and 167 females (38.6%) and had an average age of 28.32±3.087 years (age range 18-40 years), comprising 75 aged 18-25 (17.3%), 343 aged 25-33 (79.2%), 15 aged 33-40 (3.5%). Of these, 27 (6.2%) were only children, and 406 (93.8% were non-only children). Their educational level were 48 (11.1%) High school or below, 198 (45.7%) Jonior college, 175 (40.4%) Bachelor’s degree, 11 (2.5%) Master’s degree, and 1 (0.2%) doctor’s degree. Of all the participants, 37 (8.5%) worked for 0-3 years, 143 (33.0%) worked for 3-5years, and 253 (58.4%) worked for more than 5 years. Regarding marital status, 329 (76.0%) were unmarried, 103 (23.8%) were married, and one (0.2%) was divorced. 【201-210】

Regarding all the descriptive analyses and the model used, because the Manifest Variables of the structural equation model were used in the study, and the fitting index of the saturated model was reached, the descriptive correlation analysis did not include the dimensions of the questionnaire. Since the subdimensions of the questionnaire were not used, the authors believed that the correlation analysis should not add subdimensions. However, we still did the correlation analysis including the questionnaire dimension.

If necessary, we can show the following table 2 in the manuscript. We hope that your confusion are solved now. Thank you very much for the detailed comments again.

Table 2. descriptive statistics, correlations among the study variables.

Steeger, C. M., & Gondoli, D. M. (2013). Mother-adolescent conflict as a mediator between adolescent problem behaviors and maternal psychological control. Developmental Psychology, 49(4), 804–814.

Wang, Q., Xin, Z., Zhang, H., Du, J., & Wang, M. (2022). The effect of the supervisor-student relationship on academic procrastination: the chain-mediating role of academic self-efficacy and learning adaptation. International journal of environmental research and public health, 19(5), 2621.

Wen, Z. J., & Ye, B. J. (2014). Analysis of mediating effects: The development of methods and models. Advances in Psychological Science, 22(5), 731-745.

Zhang, L., Xue X. J., & Zhao, J. X. (2019). Discrimination perception, depression, and academic achievement in rural left-behind children: A longitudinal mediation model. Journal of Psychological Science, 42(3), 584-590.

  1. Authors can move in this section some sentences used in the results section related to the justification of the test used.

Response:

Thank you so much for the detailed comments.

In this section, introduction of saturation models and indicators used has been added, so some sentences in the results section have not been moved over.

A Saturated model is a condition on the assumption that all the parameters to be estimated in the model are exactly equal to the elements in the covariance matrix, the degree of freedom of the saturated model is 0, the chi-square values is also equal to 0, and the saturated model is called just identified, which leads to perfect fitting, so the fitting index is no longer estimated (Steeger & Gondoli, 2013; Wen & Ye, 2014; Zhang et al., 2019). Let’s just focus on the path coefficients. We used chi-square values (χ2/df), the comparative fit index (CFI), the Tucker–Lewis fit index (TLI), the root mean square error of approximation (RMSEA), and the standardized root mean square residual (SRMR) to evaluate the models. In general, saturated model fit is indicated by CFI and TLI is 1 and RMSEA and SRMR is 0 (Wang et al., 2022). 【244-255】

Wen, Z. J., & Ye, B. J. (2014). Analysis of mediating effects: The development of methods and models. Advances in Psychological Science, 22(5), 731-745.

Zhang, L., Xue X. J., & Zhao, J. X. (2019). Discrimination perception, depression, and academic achievement in rural left-behind children: A longitudinal mediation model. Journal of Psychological Science, 42(3), 584-590.

  1. I propose to write not only a discussion but also conclusions and future research directions.

Response:

Thank you very much for the detailed comments.

Based on your comments, the authors added limitations, future research directions, and conclusions.

4.5 Limitations and Future Directions

There are several limitations of this study that should be noted.

First, although we have made great efforts to collect sample data, future research should focus on youth living alone and expand the sample size to explore differences in psychological characteristics between youth living alone and other groups. Furthermore, the quantitative study is not enough to draw the conclusion of causality between self-concealment and perceived social support among youth living alone. In the future, the typical psychological aspects of this group should be evaluated through qualitative analysis, and studies should explore the influencing factors, development stages, and mechanism of social support among youth living alone. Last but not least, due to the Chinese culture such collectivism, marriageism, and lineage,youth living alone in China are likely to be stigmatized. In this regard, relevant questionnaires or implicit association tests can be used for cross-cultural studies in the future.

5 Conclusion

(1) self-concealment was significantly negatively correlated with psychological needs internet gratification, and was significantly positively correlated with social self-esteem and perceived social support. Psychological needs internet gratification was significantly negatively correlated with social self-esteem and perceived social support. Social self-esteem was significantly positively correlated with perceived social support.

(2) The effect of perceived social support of youth living alone included the direct effect of self-concealment and indirect effect through psychological needs internet gratification and social self-esteem. Psychological needs internet gratification and social self-esteem play a chain intermediary role between self-concealment and perceived social support of youth living alone. 【461-486】

Thank you so much for the helpful and detailed comments. We have learned much from them. We hope we have answered the questions you are concerned. If not, we would love to make a further modification.

Reviewer 2 Report

Dear Authors, 

Review for the Manuscript “Living Alone but Not Feeling Alone: The effect of Self-Concealment on Perceived Social Support of Youth Living Alone in China.”

The paper focuses on understanding the effect of self-concealment on perceived social support, the mediating role of self-esteem and the psychological needs of Internet Gratification in a population of Chinese young people living alone.

The study uses a quantitative research design to raise the mediation relations among the constructs. The research is context-specific, which can be considered a limit and a strength.

I appreciate the originality of the manuscript. I think the article can be published in this journal before major revisions.

Introduction

Paragraphs 1.1 and 1.3:

These sections concern the relation between social support, self-concealment, and the mediating role of social self-esteem. The factors related to young people living alone are described, but I suggest adding some details on the relations among these factors. It seems to be too descriptive.  Moreover, you mention social activity (measured by the Texas Social Behavior Inventory), but the reference to social behaviours (and what you mean by using social behaviours) is too restricted. 

Line 52: please rephrase the word unaccompanied.

Lines 55-56: could you explain better what you mean by “single and fixed circle of communication, little space for promotion at work, implicit discrimination”?

Line 58: could you detail the meaning of “perceiving the world in the way that they understand”?

Line 76: I am unsure if this sentence is appropriate for this paper “Healthy children perceive more 76 social support than children with ADHD (Emser & Christiansen, 2021)”. In the following lines, you indicate the age range of 18- 40, but I suggest specifying and justifying the use of this range of age, considering that your frame is young people. Do and how you consider young people when 40 years old?

Line 96: I think that the perspective of agency of young people that choose to live alone could be raised in more detail in this section.

Line 156: could you please explain what you mean by “suppositional social activity”?

Methods

I suggest describing the sample of participants by adding education level, socio-economic status and type of employment (if present).

Line 203: Social self-esteem is measured using the Texas Social behaviours Inventory. Could you explain in the literature part how you consider the two concepts and whether and how they can be measured through this sale?

Results

I suggest revising the tables and the overall description of the analysis implemented. You could also consider restructuring table 2. Can the differences between men and women be reported? And those regarding the age groups?

Line 247: Could you explain in detail what you mean by saturated model and add appropriate references that justify your affirmation? Moreover, I suggest briefly describing the type of analysis you used and the meaning of the coefficients you reported.

Discussion

I suggest carefully revising some interpretations of your results; for example, the sentence “This is an interpersonal communication strategy which increases the individual's interpersonal rejection caused by social threats (MacDonald &  Leary, 2005)”; how do you reach this conclusion given that you do not mention any social threats or risk conditions that could force young people to live alone? Moreover, I suggest integrating them into a more coherent section of the discussions.

Lines 315- 318: please rephrase this sentence. It is unclear to me.

Lines 319-322: what is the meaning of introducing the biological maturation of human beings? I do not understand. Is it helpful to justify your results? I suggest removing this sentence and reporting appropriate justifications.

Lines 328-329: Therefore, youth living alone conceal some negative information and negative emotions that they think are unnecessary and have nothing to do with others. How do you argue this result regarding the well-being of young people living alone?

Line 353: psychological need satisfaction on the Internet and in reality. What is the difference between the two? I suggest introducing it in the literature review, if any.

Line 368: The results in the present study support the theory of 368 self-protection. Could you explain the content of this theory if relevant to your results?

Line 381: the “Use It or Lose It” hypothesis. Could you explain what you refer to?

Finally, I suggest adding a section on the Limits section and one on the Chinese context, mainly referring to how young people living alone are represented and/or stigmatised in the Chinese context.

Please, check the format of tables and figures with the Journal requirements.

Author Response

Modified description

Thank you very much for the detailed reviewer’s comments and suggestions. The authors have revised and improved the whole paper (marked in blue font) based on the reviewer’s comments and suggestions. We hope that the revised manuscript meets the journal’s standards.

Reviewer 2

The paper focuses on understanding the effect of self-concealment on perceived social support, the mediating role of self-esteem and the psychological needs of Internet Gratification in a population of Chinese young people living alone.

The study uses a quantitative research design to raise the mediation relations among the constructs. The research is context-specific, which can be considered a limit and a strength.

Response:

Thank you very much for your affirmation of our hard work. Your opinions let us feel your love for academic research and meticulous research attitude again.

Introduction

Paragraphs 1.1 and 1.3:

1 These sections concern the relation between social support, self-concealment, and the mediating role of social self-esteem. The factors related to young people living alone are described, but I suggest adding some details on the relations among these factors. It seems to be too descriptive. Moreover, you mention social activity (measured by the Texas Social Behavior Inventory), but the reference to social behaviours (and what you mean by using social behaviours) is too restricted.

Response:

Thank you very much for the detailed comments.

Based on your comments, the authors have made some changes.

Living alone can also help in maintaining a good self-image, promoting harmony and stability of interpersonal relationships (Larson et al., 2015; Kelly, 2002), and gaining the psychological feeling of being supported by others. Therefore, youth living alone actively choose to live independently and conceal adverse events. 【100-104】

In China, independence is the opposite psychological character of social communication (Wang & Hu, 2022). Besides, the social emotional needs on the Internet emphasize the clustering of individuals or groups, rather than independence. (Yin & Liu, 2021). 【127-131】

When individuals do not use this strategy, they are likelier to feel out of self-control and have low social efficacy. Therefore, self-concealment can positively predict social self-esteem. Meanwhile, individuals with higher levels of social self-esteem have the more robust ability to perceived social support from others(Li et al., 2014). 【154-158】

2 Line 52: please rephrase the word unaccompanied.

Response:

Thank you very much for the detailed comments.

I’m so sorry there was a translation error in the original manuscript. We have revised this mistake. “unaccompanied youth” is should be youth living alone, and this statement has been corrected in the text.

3 Lines 55-56: could you explain better what you mean by “single and fixed circle of communication, little space for promotion at work, implicit discrimination”?

Response:

Thank you very much for the detailed comments.

Based on your comments, the authors explain in more detail.

According to previous studies, the situation of youth living alone is not optimistic, such as living in tiny houses, poor environment, high rent, and loneliness; after staying in big cities for a period, they usually interact with fellow natives, colleagues, or clients, so that their interpersonal relationship is a single and fixed form. At the same time, youth living alone may be subject to implicit discrimination at work, such as when they work overtime, they are taken for granted as the first candidates for overtime (Ma, 2019). 【53-60】

4 Line 58: could you detail the meaning of “perceiving the world in the way that they understand”?

Response:

Thank you very much for the detailed comments.

There’s a growing group of youth living alone, since-unlike family households-they feel their needs, perceive the world, understand the message addressed to them differently, have different systems of values, and exhibit different behaviours (Zalega, 2020).

Youth living alone are more independent and have higher abilities of self-knowledge, self-evaluation, self-development, and self-realization (Cui & Tan, 2020). 【104-106】

Zalega, T. (2020). Sustainable consumption in consumer behaviour of young polish singles. Acta Scientiarum Polonorum – Oeconomia19(1), 89-100.

5 Line 76: I am unsure if this sentence is appropriate for this paper “Healthy children perceive more 76 social support than children with ADHD (Emser & Christiansen, 2021)”. In the following lines, you indicate the age range of 18- 40, but I suggest specifying and justifying the use of this range of age, considering that your frame is young people. Do and how you consider young people when 40 years old?

Response:

Thank you very much for the detailed comments.

The authors agree with you on this point. “Healthy children perceive more social support than children with ADHD (Emser & Christiansen, 2021).” This sentence has been deleted.

About this range of age, China 2022 age classification criteria: minors (below 18 years old), young people (18-44 years old), middle-aged people (40-59 years old), young old people (60-74 years old), and elder people (75-89 years old).

With the development of the economy and technology, the improvement of medical level and living standard, the delay of marriage age, and the prolongation of life expectancy, combined with previous studies, the age is young people defined as 18-40. And there was only one participant aged 40 in this study. In the meantime, the age range of young subjects in other studies also included 40 (He & Tan, 2021; Luo, 2017; Yang, 2020).

Thank you so much for the helpful and detailed comments. We have learned much from them. We hope we have answered this question you are concerned. If not, we would love to make a further modification.

He, H., & Tan T. (2021). The trend of mean age at first marriage and the factor-specific contribution rate of late marriage in China. Population Journal, 43(5), 16-28.

Luo, C. (2017). Re-partitioning population age group and its implications. Population Journal, 41(5), 16-25.

Yang, Y. N. (2020). The effect of delayed marriage age and couples age gap on the birth will in China. South China Population, 35(3), 21-32.

6 Line 96: I think that the perspective of agency of young people that choose to live alone could be raised in more detail in this section.

Response:

Thank you very much for the detailed comments.

From the perspective of theory, Buchholz’s need theory indicates that being alone is a developmental need, which helps individuals to perceive and regulate their negative emotions (Buchholz & Anisfeld, 2000; Bond, 1990; Morrison, 1986). Further, the theory provides psychological and environmental space for self-healing and pain relief (Koch, 1994). 【96-100】

From the perspective of reality, living alone can also help in maintaining a good self-image, promoting harmony and stability of interpersonal relationships (Larson et al., 2015; Kelly, 2002), and gaining the psychological feeling of being supported by others. 【100-103】

Therefore, youth living alone actively choose to live independently and conceal adverse events.

7 Line 156: could you please explain what you mean by “suppositional social activity”?

Response:

Thank you very much for the detailed comments.

I’m so sorry there was a translation error in the original manuscript. We have revised this mistake. The correct statement is “Displacing Social Activity”.

This theory states that the time spent by individuals using the Internet replaces the time spent on real social activities (Kraut et al., 1998). Similar to other negative non-social entertainment activities, the Internet provides personal entertainment activities, and people get a certain sense of pleasure from this type of entertainment. These entertainment activities can result in withdrawal from social relationships and reduce the personal sense of self-worth in social communication. 【Line 165-171】

Methods

8 I suggest describing the sample of participants by adding education level, socio-economic status and type of employment (if present).

Response:

Thank you very much for the detailed comments.

Based on your comment, the authors added more information about the characteristics of the sample (marital status, educational level, worked years).

The subjects included in this study comprised 266 males (61.4%) and 167 females (38.6%) and had an average age of 28.32±3.087 years (age range 18-40 years), comprising 75 aged 18-25 (17.3%), 343 aged 25-33 (79.2%), 15 aged 33-40 (3.5%). Of these, 27 (6.2%) were only children, and 406 (93.8% were non-only children). Their educational level were 48 (11.1%) High school or below, 198 (45.7%) Jonior college, 175 (40.4%) Bachelor’s degree, 11 (2.5%) Master’s degree, and 1 (0.2%) doctor’s degree. Of all the participants, 37 (8.5%) worked for 0-3 years, 143 (33.0%) worked for 3-5years, and 253 (58.4%) worked for more than 5 years. Regarding marital status, 329 (76.0%) were unmarried, 103 (23.8%) were married, and one (0.2%) was divorced. 【201-210】

9 Line 203: Social self-esteem is measured using the Texas Social behaviours Inventory. Could you explain in the literature part how you consider the two concepts and whether and how they can be measured through this sale?

Response:

Thank you very much for the detailed comments.

In this study, the authors used the Texas Social Behaviors Inventory to measure social self-esteem, which was selected according to previous studies. The references are as follows.

Guo, S., Cheng, Z. H., Liu, X. M., & Hu, H. (2015). Relationship of doctor-patient communication with shyness and social self-esteem in medical interns. Chinese journal of clinical psychology, 23(5), 812-814.

Li, Y. J., Zhu, X. S., & Chen, Y. H. (2014). The relationship of college students’ social support and well-being: the mediating role of social self-esteem. Studies of psychology and behavior, 12(3), 351-356.

Pritchard, F. M. (2012). Relationships between self-esteem, media influence and drive for thinness. Eating Behaviors, 13, 321-325.

Results

10 I suggest revising the tables and the overall description of the analysis implemented. You could also consider restructuring table 2. Can the differences between men and women be reported? And those regarding the age groups?

Response:

Thank you very much for the detailed comments.

Based on your comment, the authors reviewed relevant literature, increased Boot SE and the percentage of effect size, and modified Table2 as follows.

Table 2. Bootstrap analysis of the significance test of the intermediary effect of the chain intermediary model.

Effect

Boot SE

Bootstrap 1000 times 95% CI

Percentage

Percentile

Bias corrected

Boot LLCI

Boot ULCI

Boot  LLCI

Boot ULCI

Indirect 1

0.097

0.050

0.019

0.209

0.023

0.220

29.40%

Indirect 2

0.172

0.048

0.078

0.269

0.086

0.276

52.12%

Indirect 3

0.057

0.030

0.014

0.123

0.015

0.132

17.27%

Total Indirect Effect

0.326

0.075

0.188

0.482

0.194

0.490

98.79%

Direct

0.004

0.086

-0.157

0.178

-0.171

0.162

1.21%

Total Effect

0.330

0.089

0.155

0.502

0.146

0.496

The authors concluded that the mediating effect analysis mainly studied the relationship between variables, rather than differences between gender and age groups. We also compared gender and age groups differences among variables. If necessary, we can show the following tables in the manuscript. We hope that your confusion are solved now. Thank you very much for the detailed comments again.

Table 3 Gender differences in all variables.

Gender

Number

M±SD

t

p

Self-concealment

male

266

3.306±0.560

1.348

0.178

female

167

3.228±0.624

Psychological needs internet
gratification

male

266

2.328±0.309

0.062

0.951

female

167

2.326±0.341

Social self-esteem

male

266

3.193±0.256

2.823

0.005

female

167

3.119±0.286

Perceived social support

male

266

4.553±0.648

0.685

0.493

female

167

4.508±0.697

Table 4 Age groups differences in all variables.

Age groups

(Years)

Number

M±SD

F

P

Self-concealment

18-25

75

3.049±0.736

8.342

0.000

25-33

343

3.334±0.537

33-40

15

3.087±0.541

Psychological needs internet
gratification

18-25

75

2.387±0.432

4.059

0.018

25-33

343

2.307±0.289

33-40

15

2.327±0.321

Social self-esteem

18-25

75

3.075±0.347

5.507

0.004

25-33

343

3.186±0.248

33-40

15

3.121±0.244

Perceived social support

18-25

75

4.478±0.877

1.9

0.151

25-33

343

4.560±0.608

33-40

15

4.250±0.709

11 Line 247: Could you explain in detail what you mean by saturated model and add appropriate references that justify your affirmation? Moreover, I suggest briefly describing the type of analysis you used and the meaning of the coefficients you reported.

Response:

Thank you very much for the detailed comments.

A Saturated model is a condition on the assumption that all the parameters to be estimated in the model are exactly equal to the elements in the covariance matrix, the degree of freedom of the saturated model is 0, the chi-square values is also equal to 0, and the saturated model is called just identified, which leads to perfect fitting, so the fitting index is no longer estimated (Steeger & Gondoli, 2013; Wen & Ye, 2014; Zhang et al., 2019). Let’s just focus on the path coefficients. We used chi-square values (χ2/df), the comparative fit index (CFI), the Tucker–Lewis fit index (TLI), the root mean square error of approximation (RMSEA), and the standardized root mean square residual (SRMR) to evaluate the models. In general, saturated model fit is indicated by CFI and TLI is 1 and RMSEA and SRMR is 0 (Wang et al., 2022). 【244-255】

Steeger, C. M., & Gondoli, D. M. (2013). Mother-adolescent conflict as a mediator between adolescent problem behaviors and maternal psychological control. Developmental Psychology, 49(4), 804–814.

Wang, Q., Xin, Z., Zhang, H., Du, J., & Wang, M. (2022). The effect of the supervisor-student relationship on academic procrastination: the chain-mediating role of academic self-efficacy and learning adaptation. International journal of environmental research and public health, 19(5), 2621.

Wen, Z. J., & Ye, B. J. (2014). Analysis of mediating effects: The development of methods and models. Advances in Psychological Science, 22(5), 731-745.

Zhang, L., Xue X. J., & Zhao, J. X. (2019). Discrimination perception, depression, and academic achievement in rural left-behind children: A longitudinal mediation model. Journal of Psychological Science, 42(3), 584-590.

Discussion

12 I suggest carefully revising some interpretations of your results; for example, the sentence “This is an interpersonal communication strategy which increases the individual's interpersonal rejection caused by social threats (MacDonald & Leary, 2005)”; how do you reach this conclusion given that you do not mention any social threats or risk conditions that could force young people to live alone? Moreover, I suggest integrating them into a more coherent section of the discussions.

Response:

Thank you very much for the detailed comments.

The authors agree with you very much, based on your comments, we have modified this sentence to read, “this is an interpersonal communication (MacDonald & Leary, 2005)”.

To be specific, youth living alone actively conceal some information about themselves owing to the accumulation of work experience and the complexity of interpersonal communication. This is an interpersonal communication strategy (MacDonald & Leary, 2005).

We have addressed similar issues in full.

MacDonald, G., & Leary, M. R. (2005). Why does social exclusion hurt? The relationship between social and physical pain. Psychological Bulletin, 131(2), 202–223.

13 Lines 315- 318: please rephrase this sentence. It is unclear to me.

Response:

Thank you very much for the detailed comments.

Based on your comments, the authors modified it as follows:

The reason for the inconsistency with the results of this study may be that the subjects of previous studies were mainly high school students and college students, but not youth living alone. 【352-353】

14 Lines 319-322: what is the meaning of introducing the biological maturation of human beings? I do not understand. Is it helpful to justify your results? I suggest removing this sentence and reporting appropriate justifications.

Response:

Thank you very much for the detailed comments.

Based on your comments, the authors removed this sentence.

To be specific, the reason for the inconsistent results may be that the subjects of previous studies were mainly high school students and college students, but not youth living alone. Youth living alone are employed and financially independent. Therefore, their social cognition level and experience are higher compared with those of high school and college students. 【352-356】

15 Lines 328-329: Therefore, youth living alone conceal some negative information and negative emotions that they think are unnecessary and have nothing to do with others. How do you argue this result regarding the well-being of young people living alone?

Response:

Thank you very much for the detailed comments. The reasons of this question as follows:

(1) According to a survey reported by the People's Think Tank, the reasons for living alone are as follows: (i) they want their privacy; (ii) they have a different life schedule from others; (iii) they live alone easily; (iv) they passively chose to live alone because their roommate moved away (Yin, 2020). 【lines 48-52】

(2) Actually, they merely empty the living space, but not the social space (He, 2017). A higher proportion of youth living alone is highly educated than those not living alone (Zhang, & Wei, 2022). They have a strong sense of independence. Meanwhile, they also have more self-respect and self-confidence (He, 2017). They believe that living alone is a necessary stage of life (Dou, 2019). 【lines 63-68】

(3) Krinenberg (2018) reported that living alone does not always results in loneliness and may promote self-attention and self-reflection in youth living alone. Notably, their attention to the outside world and others is relatively weakened. 【lines 83-86】

(4) Moreover, Maslow reported that being alone is one of the characteristics of self-actualizers, and can help in realizing the benefits of being alone such as enjoy being alone, and to feel relaxed and comfortable (Maslow, 1970). 【lines 86-89】

(5) Youth living alone are more independent and have higher abilities of self-knowledge, self-evaluation, self-development, and self-realization (Cui & Tan, 2020). Negative information from the outside and from other individuals has less impact on this group, and they prefer to solve personal problems by themselves and maintain privacy (Zhang, 2020). 【lines 104-108】

To sum up, youth living alone merely empty the living space, but not the social space. They are relatively well-educated, more independent, and have higher abilities of self-knowledge, self-evaluation, self-development, and self-realization.

16 Line 353: psychological need satisfaction on the Internet and in reality. What is the difference between the two? I suggest introducing it in the literature review, if any.

Response:

Thank you very much for the detailed comments. The explanations for this question are as follows:

(1) Use of the Internet mainly satisfies people’s need for emotional cathartic, social communication, entertainment, and professional status identification and other needs (Ferguson & Perse, 2000; LaRose et al., 2001; Parker & Plank, 2000; Wolin, 1999). 【lines 120-123】

(2) Moreover, the Anonymity-Convenience-Escapism (ACE) model proposes that Internet can satisfy the need for individual anonymity, convenience, and escape from reality (Young, 2004). 【lines 137-139】

(3) The hyper-interpersonal model proposed by Walther (1996) indicates that individuals perform hyper-interpersonal communication in online interpersonal communication, which is different from general face-to-face interpersonal communication. The sender and receiver of information selectively present themselves in the process of hyper-interpersonal communication. 【lines 370-373】

17 Line 368: The results in the present study support the theory of self-protection. Could you explain the content of this theory if relevant to your results?

Response:

Thank you very much for the detailed comments. The explanations for this question are as follows:

(1) Previous studies indicate that self-protection is a defensive strategy, and self-protection can help in avoiding, reducing and repairing negative self-perceptions caused by evaluations from other individuals (Hepper et al., 2010). 【lines 147-149】

(2) This indicates that the self-worth and social competence of individuals increases when they use self-protection strategies to conceal themselves in social interactions. When individuals do not use this strategy, they are likelier to feel out of self-control and have low social efficacy. Therefore, self-concealment can positively predict social self-esteem. 【lines 152-156】

The results of the present study support the theory of self-protection.

18 Line 381: the “Use It or Lose It” hypothesis. Could you explain what you refer to?

Response:

Thank you very much for the detailed comments.

Midkiff (2004) thought that research suggests that there is a “use it or lose it” component on cognitive tasks and that performance can be moderated by an individual’s exposure to complex stimulus in his/her environment.

The “Use It or Lose It” hypothesis indicates that frequent social activities increase cognitive activity of an the individual and increase sensitivity of social feedback. This sensitivity gradually decreases with absence of social activities (Krueger et al., 2009; Midkiff, 2004). 【414-417】

According to the authors, it refers to the knowledge or skills we have, which will decline or disappear after a long time without use. Frequent social activities will increase the cognitive activity of individuals and make them more likely to feel support from social objects (friends, family, etc.). Therefore, individuals with high social self-esteem may feel higher social support.

Midkiff, K. (2004). Use it or lose it? What predicts age-related declines in cognitive performance in elderly adults?. McNair Scholars Journal, 8(1), 53-59.

19 Finally, I suggest adding a section on the Limits section and one on the Chinese context, mainly referring to how young people living alone are represented and/or stigmatised in the Chinese context.

Response:

Thank you very much for the detailed comments.

Based on your comments, the authors added limitations, future research directions.

4.5 Limitations and Future Directions

There are several limitations of this study that should be noted.

First, although we have made great efforts to collect sample data, future research should focus on youth living alone and expand the sample size to explore differences in psychological characteristics between youth living alone and other groups. Furthermore, the quantitative study is not enough to draw the conclusion of causality between self-concealment and perceived social support among youth living alone. In the future, the typical psychological aspects of this group should be evaluated through qualitative analysis, and studies should explore the influencing factors, development stages, and mechanism of social support among youth living alone. Last but not least, due to the Chinese culture such collectivism, marriageism, and lineage,youth living alone in China are likely to be stigmatized. In this regard, relevant questionnaires or implicit association tests can be used for cross-cultural studies in the future. 【461-474】

20 Please, check the format of tables and figures with the Journal requirements.

Response:

Thank you very much for the detailed comments.

The authors modified the format of tables and figures according to the Journal requirements.

Thank you so much for the helpful and detailed comments. We have learned much from them. We hope we have answered the questions you are concerned. If not, we would love to make a further modification.

Reviewer 3 Report

To the authors,

The idea of having explored the mechanism of self-concealment on perceived social support among youth living alone is very suggestive and contributes to consolidate the cumulative knowledge about a growing phenomenon. Moreover, the proposal of a causal model that includes mediating variables such as psychological needs, internet gratification and social self-esteem, which are especially relevant in youth samples, contributes to shedding light on the dynamics of influence between self-concealment on perceived social support.

The paper is well structured and easy-to-follow. Extensive and well-articulated introduction, clearly described assessment instruments, objectives-methodology-results-discussion in clear agreement, are some of the strengths. 

I have only few comments that I hope could be helpful in improving the manuscript.

·     In the introduction, we recommend that the authors comment in greater detail on the objectives of the study in the paragraph including lines 65-72. Thus, it would make more sense to include subsections 1.1, 1.2, 1.3, and 1.4.

  It would be desirable that the authors, in subsection 2.1. Participants and procedure, provide more information about the sample collection procedure.

·       Do the authors have more information (beyond age and gender) about the characteristics of the sample (e.g., marital status, educational level, type of work performed, etc.), if so, this should be reported.

·       In the discussion, we suggest that the authors elaborate an initial paragraph prior to subsections 4.1, 4.2, 4.3 and 4.4 to bring the potential reader closer to the different aspects analyzed in the paper (probably lines 296-303 would fit better in this new paragraph).

·       It seems appropriate to include a conclusions section

·       We recommend including the strengths and limitations of the study and advancing the main practical implications of the results obtained with this research.

Author Response

Modified description

Thank you very much for the detailed reviewer’s comments and suggestions. The authors have revised and improved the whole paper (marked in blue font) based on the reviewer’s comments and suggestions. We hope that the revised manuscript meets the journal’s standards.

Reviewer 3

The idea of having explored the mechanism of self-concealment on perceived social support among youth living alone is very suggestive and contributes to consolidate the cumulative knowledge about a growing phenomenon. Moreover, the proposal of a causal model that includes mediating variables such as psychological needs, internet gratification and social self-esteem, which are especially relevant in youth samples, contributes to shedding light on the dynamics of influence between self-concealment on perceived social support.

The paper is well structured and easy-to-follow. Extensive and well-articulated introduction, clearly described assessment instruments, objectives-methodology-results-discussion in clear agreement, are some of the strengths.

Response:

Thank you very much for your affirmation of our hard work. Your opinions let us feel your love for academic research and meticulous research attitude again.

1 In the introduction, we recommend that the authors comment in greater detail on the objectives of the study in the paragraph including lines 65-72. Thus, it would make more sense to include subsections 1.1, 1.2, 1.3, and 1.4.

Response:

Thank you very much for the detailed comments.

Based on your comments, the authors have made some changes.

This study uses quantitative research to explore the influence mechanism of self-concealment on perceived social support among youth living alone and Chain mediating effect of psychological needs internet gratification and social self-esteem. So that individuals, families, and society can access to an in-depth understanding of the status quo and characteristics about the social behavior of youth living alone. 【70-75】

2 It would be desirable that the authors, in subsection 2.1. Participants and procedure, provide more information about the sample collection procedure.

Response:

Thank you very much for the detailed comments.

Based on your comments, the authors added more information about the sample collection procedure in subsection 2.1 Participants and procedure.

Firstly,the Questionnaire Star software was used for online recruitment, and 78 eligible participants were recruited. Then, the purposive sampling method was used for re-recruitment, and 135 eligible participants were recruited. Finally, based on purposive sampling, the snowball sampling was used, and 180 eligible participants were recruited. 【197-201】

3 Do the authors have more information (beyond age and gender) about the characteristics of the sample (e.g., marital status, educational level, type of work performed, etc.), if so, this should be reported.

Response:

Thank you very much for the detailed comments.

Based on your comments, the authors added more information about the characteristics of the sample (marital status, educational level, worked years).

The subjects included in this study comprised 266 males (61.4%) and 167 females (38.6%) and had an average age of 28.32±3.087 years (age range 18-40 years), comprising 75 aged 18-25 (17.3%), 343 aged 25-33 (79.2%), 15 aged 33-40 (3.5%). Of these, 27 (6.2%) were only children, and 406 (93.8% were non-only children). Their educational level were 48 (11.1%) High school or below, 198 (45.7%) Jonior college, 175 (40.4%) Bachelor’s degree, 11 (2.5%) Master’s degree, and 1 (0.2%) doctor’s degree. Of all the participants, 37 (8.5%) worked for 0-3 years, 143 (33.0%) worked for 3-5years, and 253 (58.4%) worked for more than 5 years. Regarding marital status, 329 (76.0%) were unmarried, 103 (23.8%) were married, and one (0.2%) was divorced. 【201-210】

4 In the discussion, we suggest that the authors elaborate an initial paragraph prior to subsections 4.1, 4.2, 4.3 and 4.4 to bring the potential reader closer to the different aspects analyzed in the paper (probably lines 296-303 would fit better in this new paragraph).

Response:

Thank you very much for the detailed comments.

This comment was so good that the authors adjusted it and created a new paragraph from lines 296-303 before 4.1, 4.2, 4.3 and 4.4.

The present study sought to explore the relationship between self-concealment and perceived social support of youth living alone, by examining the mediating role of psychological needs internet gratification and social self-esteem. The results from mediation analyses provided information on potential mechanisms that explain the relationship between self-concealment and perceived social support. Three types of mediation effects were observed, with psychological needs internet gratification and social self-esteem as separate single mediators and these variables playing a role as serial mediators. 【332-339】

5 It seems appropriate to include a conclusions section

Response:

Thank you very much for the detailed comments.

Based on your comments, the authors added conclusions.

5 Conclusion

(1) self-concealment was significantly negatively correlated with psychological needs internet gratification, and was significantly positively correlated with social self-esteem and perceived social support. Psychological needs internet gratification was significantly negatively correlated with social self-esteem and perceived social support. Social self-esteem was significantly positively correlated with perceived social support.

(2) The effect of perceived social support of youth living alone included the direct effect of self-concealment and indirect effect through psychological needs internet gratification and social self-esteem. Psychological needs internet gratification and social self-esteem play a chain intermediary role between self-concealment and perceived social support of youth living alone. 【475-486】

6 We recommend including the strengths and limitations of the study and advancing the main practical implications of the results obtained with this research.

Response:

Thank you very much for the detailed comments.

Based on your comments, the authors added limitations, future research directions.

4.5 Limitations and Future Directions

There are several limitations of this study that should be noted.

First, although we have made great efforts to collect sample data, future research should focus on youth living alone and expand the sample size to explore differences in psychological characteristics between youth living alone and other groups. Furthermore, the quantitative study is not enough to draw the conclusion of causality between self-concealment and perceived social support among youth living alone. In the future, the typical psychological aspects of this group should be evaluated through qualitative analysis, and studies should explore the influencing factors, development stages, and mechanism of social support among youth living alone. Last but not least, due to the Chinese culture such collectivism, marriageism, and lineage,youth living alone in China are likely to be stigmatized. In this regard, relevant questionnaires or implicit association tests can be used for cross-cultural studies in the future. 【461-474】

Although youth living alone may encounter problems in dealing with insufficient urban public space development, recessive social rejection, stereotypes, and discrimination, such as label effect of adverse reactions (Zhou, 2020). However, the fact is that unaccompanied youth living alone were not confined to the surface of this sadness. It is a short-term option for a personal journey. More than anything else, they are willing to enjoy living alone and learn responsibility. They can arrange their time wisely to make constant progress and improve themselves. Enduring solitude allows them to gain control over their lives to pursue their dreams with all their might (Dou, 2019). 【452-460】

Thank you so much for the helpful and detailed comments. We have learned much from them. We hope we have answered the questions you are concerned. If not, we would love to make a further modification.

Round 2

Reviewer 2 Report

Dear authors, I appreciate your effort in reviewing the manuscript. 

In my opinion, some final improvements are still needed:

- regarding the difference in age and gender, could you please report and argue the relevant differences in the results and discussion sections?;

- regarding the added sentence, "In China, independence is the opposite psychological character of social communication (Wang & Hu, 2022)", could you please argue what independence means in the Chinese context? Is it valued positively? Why should it be the opposite of social communication? Do you mean that independent men/women are not social?

- revise the editing of the conclusion section to make it more readible

Kind regards

Author Response

Thank you very much for the detailed reviewer’s comments and suggestions. The authors have revised and improved the whole paper (marked in green font) based on the reviewer’s comments and suggestions. We hope that the revised manuscript meets the journal’s standards.

Reviewer 2

Dear authors, I appreciate your effort in reviewing the manuscript.

Response:

Thank you very much for your affirmation of our hard work. Your opinions let us feel your love for academic research and meticulous research attitude again.

1 Regarding the difference in age and gender, could you please report and argue the relevant differences in the results and discussion sections?

Response:

Thank you very much for the detailed comments.

Based on your comments, the authors added more information in the results and discussion sections.

3.2 Preliminary Analysis

Compared to females, males scored higher on average in all variables. Independent sample t-tests revealed no significant differences between the sexes regarding self-concealment, psychological needs internet gratification, and perceived social support. A significant sex difference was found regarding social self-esteem.

Table 1 Gender differences in all variables.

Gender

Number

M±SD

t

p

Self-concealment

male

266

3.306±0.560

1.348

0.178

female

167

3.228±0.624

Psychological needs internet
gratification

male

266

2.328±0.309

0.062

0.951

female

167

2.326±0.341

Social self-esteem

male

266

3.193±0.256

2.823

0.005

female

167

3.119±0.286

Perceived social support

male

266

4.553±0.648

0.685

0.493

female

167

4.508±0.697

ANOVAs showed no significant interaction effect between gender and age group on the variables, i.e., self-concealment, psychological needs internet gratification, social self-esteem, and perceived social support. Sample main effect analysis showed that: (i) self-concealment: where youth living alone aged 25-33years (3.334±0.537) reported higher self-concealment than 18-25years (3.049±0.736) (p<0.001); (ii) psychological needs internet gratification: where youth living alone aged 25-33 years (2.307±0.289) had a lower psychological needs internet gratification, compared to aged 18-25years (2.387±0.432) (p<0.05) and 33-40 years (2.327±0.321) (p<0.05); (iii) social self-esteem: in which youth living alone aged 25-33years (3.186±0.248) reported higher social self-esteem than 18-25years (3.075±0.347) (p<0.001). 【Line 267-291】

Table 2 Age groups differences in all variables.

Age groups (Years)

Number

M±SD

F

P

Self-concealment

①18-25

75

3.049±0.736

8.342

0.001

②25-33

343

3.334±0.537

③33-40

15

3.087±0.541

Psychological needs internet
gratification

①18-25

75

2.387±0.432

4.059

0.018

②25-33

343

2.307±0.289

③33-40

15

2.327±0.321

Social self-esteem

①18-25

75

3.075±0.347

5.507

0.004

②25-33

343

3.186±0.248

③33-40

15

3.121±0.244

Perceived social support

①18-25

75

4.478±0.877

1.900

0.151

②25-33

343

4.560±0.608

③33-40

15

4.250±0.709

4.1. The difference in age and gender

This study found that males living alone had significantly higher levels of social self-esteem than females, which was consistent with previous studies (Luo & Chen, 2009). The result showed that under the influence of traditional Chinese culture, males paid more attention to social ties than females, and pursued personal success and the realization of self-worth (Li et al., 2007).

Regarding age, the degree of self-concealment of youth living alone aged 25-33 was significantly higher than that of 18-25 years old. The possible reason for this result is that youth living alone aged 25-33 have a higher level of social cognition and pay more attention to self-growth. The negative information of the outside world has less influence on them, so they are more inclined to self-concealment. The level of psychological needs internet gratification of youth living alone aged 25-33 was significantly lower than 18-25 and 33-40 years old. This result showed that, during the critical period of work and career growth, youth living alone were more concerned about their development and did not rely too much on the Internet for emotional expression or escapism. In terms of social self-esteem, only youth living alone aged 18-25 scored significantly lower than those aged 25-33, which may be due to the fact that with the accumulation of work practice and life experience, youth living alone had improved their sense of self-worth in the social process, and their social ability had reached a certain level, and they may be in a more satisfactory state of social self-esteem. 【Line 373-393】

Berardelli, I., Rogante, E., Sarubbi, S., Erbuto, D., Cifrodelli, M., Concolato, C., ... & Pompili, M. (2022). Is Lethality Different between Males and Females? Clinical and Gender Differences in Inpatient Suicide Attempters. International Journal of Environmental Research and Public Health,19(20), 13309.

Motevalli, M., Tanous, D., Wirnitzer, G., Leitzmann, C., Rosemann, T., Knechtle, B., & Wirnitzer, K. (2022). Sex differences in racing history of recreational 10 km to ultra runners (Part B)—Results from the NURMI Study (Step 2). International Journal of Environmental Research and Public Health, 19(20), 13291.

Wang, Q., Xin, Z., Zhang, H., Du, J., & Wang, M. (2022). The effect of the supervisor-student relationship on academic procrastination: the chain-mediating role of academic self-efficacy and learning adaptation. International journal of environmental research and public health, 19(5), 2621.

Wong, N., Gong, X., & Fung, H. H. (2020). Does valuing happiness enhance subjective well-being? The age-differential effect of interdependence. Journal of Happiness Studies, 21(1), 1-14.

Luo, L. F., & Chen, M. H. (2009). The relationship among social self-esteem, family cohesion and adaptability of college students. China Journal of Health Psychology, 17(1), 43-45.

Li, J. B., Situ, Q. M., & Tao, J. (2007). Correlation study of family state and self-esteem level of freshman. China Journal of Health Psychology, 15(8), 741-743.

2 Regarding the added sentence, "In China, independence is the opposite psychological character of social communication (Wang & Hu, 2022)", could you please argue what independence means in the Chinese context? Is it valued positively? Why should it be the opposite of social communication? Do you mean that independent men/women are not social?

Response:

Thank you very much for the detailed comments.

The authors' description of the sentence is not clear enough, making the sentence ambiguous. In the context of Chinese culture, youth living alone are not unsocial, but obtain appropriate emotional expression through psychological boundaries. The authors have revised the sentence.

In China, youth living alone pay attention to the relative independence of their inner world, carry out appropriate emotional expression in interpersonal interaction, and are more willing to maintain a sense of boundary with the outside world (Wang & Hu, 2022). 【Line 127-131】

3 Revise the editing of the conclusion section to make it more readable.

Response:

Thank you very much for the detailed comments.

Based on your comments, the authors revised the conclusion section to make it more readable.

5 Conclusion

(1) In terms of gender, there is a significant difference in self-concealment among youth living alone, and male is higher than female. In terms of age, there are significant differences in self-concealment, psychological needs internet gratification, and social self-esteem, but there is no significant difference in perceived social support.

(2) Self-concealment can directly predict the perceived social support of youth living alone, and there is a significant positive correlation between them.

(3) Self-concealment can also indirectly and positively predict perceived social support of youth living alone through the mediating effects of psychological needs internet gratification, and social self-esteem. Psychological needs internet gratification and social self-esteem can separately affect this relationship, but they can also influence this relationship as linked mediators. 【Line 529-540】

Thank you so much for the helpful and detailed comments. We have learned much from them. We hope we have answered the questions you are concerned. If not, we would love to make a further modification.